**Cite this article:** ten Brink H, Seehausen O. 2022 Competition among small individuals hinders adaptive radiation despite ecological opportunity. *Proc. R. Soc. B* **289**: 20212655. https://doi.org/10.1098/rspb.2021.2655

theoretical biology, ecology, evolution

adaptive radiation, speciation, ontogenetic diet shifts, size structure, life history

**Author for correspondence:**
Hanna ten Brink
e-mail: hannatenbrink@gmail.com

# Competition among small individuals hinders adaptive radiation despite ecological opportunity

Hanna ten Brink[1] and Ole Seehausen[1,2]

[1]Eawag Swiss Federal Institute of Aquatic Science and Technology, Department of Fish Ecology and Evolution, Center of Ecology, Evolution, and Biogeochemistry, Kastanienbaum, Switzerland
[2]Division of Aquatic Ecology and Evolution, Institute of Ecology and Evolution, University of Bern, Bern, Switzerland

HtB, 0000-0001-6772-7229; OS, 0000-0001-6598-1434

Ontogenetic diet shifts, where individuals change their resource use during development, are the rule rather than the exception in the animal world. Here, we aim to understand how such changes in diet during development affect the conditions for an adaptive radiation in the presence of ecological opportunity. We use a size-structured consumer–resource model and the adaptive dynamics approach to study the ecological conditions for speciation. We assume that small individuals all feed on a shared resource. Large individuals, on the other hand, have access to multiple food sources on which they can specialize. We find that competition among small individuals can hinder an adaptive radiation to unfold, despite plenty of ecological opportunity for large individuals. When small individuals experience strong competition for food, they grow slowly and only a few individuals are recruited to the larger size classes. Hence, competition for food among large individuals is weak and there is therefore no disruptive selection. In addition, initial conditions determine if an adaptive radiation occurs or not. A consumer population initially dominated by small individuals will not radiate. On the other hand, a population initially dominated by large individuals may undergo adaptive radiation and diversify into multiple species.

## 1. Introduction

Adaptive radiations, where a lineage diversifies into multiple species exploiting a variety of niches, are responsible for much of the diversity of life [1]. To explain the origins and structure of biodiversity, it is therefore of fundamental importance to understand why some lineages undergo adaptive radiations while others do not.

A prerequisite for adaptive radiation is ecological opportunity, the availability of a multitude of resources that are unexploited by competing species [1–3]. Ecological opportunities can arise due to the appearance of new resources, the colonization of an underused area by the ancestral species, the extinction of species previously using these resources or due to key innovations providing the capability of taking advantage of available but previously inaccessible resources [3]. In the presence of ecological opportunity, individuals may be exposed to new selection regimes leading to trait diversification. Intraspecific resource competition can then induce disruptive selection on trait variation by causing negative frequency-dependent interactions. This may ultimately result in speciation in case the mating system evolves such that reproductive isolation occurs between populations with divergent phenotypes [4,5].

Current theory on the conditions for speciation via intraspecific resource competition has not taken into account the fact that individuals grow during

their life and that their ecological role changes profoundly during development (but see [6]). After an individual is born, it needs to grow before it is able to reproduce. In addition, most animal species change their diet during their development [7,8]. These ontogenetic diet shifts are often accompanied by metamorphosis, where individuals abruptly change their morphology, habitat and physiology. In species without metamorphosis, ontogenetic diet shifts are also a common phenomenon. Many fish species [9] and reptiles [7], for example, change their diet during their lifetime without undergoing abrupt changes in their morphology.

It is essential to take individual development into account when studying the conditions for speciation. Theory predicts that the strength of ecological interactions between individuals and their environment (e.g. competitors, predators and prey availability) is an important determinant of whether evolutionary diversification occurs or not [10]. Since these ecological interactions change during development, development has the potential to affect the conditions for speciation. Predation pressure or artificial harvesting, for example, is often size-specific [11] and can drive evolutionary diversification in maturation size [12,13].

Here, we aim to understand how ontogenetic diet shifts affect the conditions for adaptive radiation in the presence of ecological opportunity. To do so, we use a size-structured consumer–resource model where individual growth and reproduction depend on resource intake. We assume that newborn individuals are limited to feed upon a shared resource only (e.g. small zooplankton that many juvenile fish feed on). Later in life individuals can diversify into multiple specialized forms that are adapted to feed on specific resources. This assumption reflects the observation that in many fish species, diets are often similar during the larval period. Only later in life juveniles and adults diverge into a broad spectrum of feeding strategies [9,14,15]. To understand the ecological conditions for speciation, we make use of the adaptive dynamics framework [16]. Adaptive dynamics is a mathematical tool to study long-term evolutionary dynamics in an ecological setting. For speciation to occur, reproductive isolating mechanisms that restrict or prevent gene exchange between newly arising morphs need to evolve. To study if assortative mating may evolve in sexually reproducing individuals, we use an individual-based model, where we incorporate multi-locus genetics to describe sexual reproduction.

## 2. Model description

For speciation with gene flow to occur in sexual populations, two processes must unfold. First, the ecological conditions must result in disruptive selection. Second, assortative mating needs to evolve. We use the framework of adaptive dynamics [16] to study which ecological conditions lead to disruptive selection. To study if disruptive selection also leads to assortative mating in sexual populations, we make use of an individual-based model. For both analyses, we use the food web approach for size-structured populations proposed by Hartvig et al. [17]. In this framework, size at maturation (denoted by $m_{mat}$) together with a species niche trait value $x$ affecting resource specialization characterizes each consumer species. All model parameters are made species independent through scaling with individual body

mass $m$ and body mass at maturation $m_{mat}$. Parameter values are determined from cross-species analysis of fish communities [17] and given in the electronic supplementary material, table A1.

The variables and individual-level model equations are summarized in tables A1 and A2 of the electronic supplementary material. The ecological dynamics at the population level are given in the electronic supplementary material, table A3.

In contrast to most diversification models, we model competition for a variety of discrete resources (see also [18]). We assume that all individuals start their lives feeding on a shared resource $R_s$. Individuals undergo a discrete diet shift at a body mass of $m_{shift}$ (see electronic supplementary material, appendix B for a relaxation of this assumption) at which they get access to six well-mixed resources with densities $R_j$ ($j = 1, \ldots, 6$). There are $N$ consumer species with density $N_i$ each ($i = 1, \ldots, N$). All evolutionary time series start with a single monomorphic consumer species $N_1$ with niche trait value $x_1$.

The efficiency of food consumption of an individual of species $i$ depends on its current size $m$, its niche trait value $x_i$ and on the densities of the resources $R$, where $R = (R_s, R_1, \ldots, R_6)$ denotes the resource vector. The size-specific attack rate of species $i$ on resource $R_j$ allometrically increases with body mass $m$ following

$$a_{i,j}(m, x_i, \theta_j) = A_{i,j}(x_i, \theta_j)m^q, \tag{2.1}$$

where parameter $q$ is a positive exponent signifying that larger individuals search a larger volume per unit time. Variable $A_{i,j}(x_i, \theta_j)$ equals the trait-dependent attack rate coefficient and its value depends on the resource $j$ and the niche trait $x_i$ of the individual. We assume that for each resource, there exists an optimal niche trait value which maximizes the attack rate on this resource. When the feeding niche trait $x_i$ of an individual equals $\theta_j$, the coefficient $A_{i,j}(x_i, \theta_j)$ on resource $j$ equals the maximum value $A_{max}$. This trait-dependent attack rate coefficient decreases in a Gaussian manner as $x_i$ moves away from the optimal trait value $\theta_j$ following

$$A_{i,j}(x_i, \theta_j) = A_{max}\exp\left[\frac{-(x_i - \theta_j)^2}{(2\tau_j^2)}\right]. \tag{2.2}$$

In the equation, parameter $\tau_j$ determines the width of the Gaussian function.

We assume two different life histories for the consumers. (i) Individuals do not need special adaptations to feed upon the shared resource when small ($\tau_s = \infty$ such that $A_{i,s}(x_i, \theta_s) = A_{max}$ for all values of $x_i$.). (ii) Alternatively, we assume a developmental trade-off between feeding on the shared resource and the other resources. Individuals highly specialized on the shared resource as juveniles (i.e. when $x_i$ is close to $\theta_s$) are less efficient as adults in feeding upon resources that differ strongly from the shared resource ($\tau_s = 20$ such that $A_{i,s}(x_i, \theta_s) < A_{max}$ when $x_i \neq \theta_s$). The feeding curves for all other resources (equation (2.2)) have a width of $\tau_j = \tau = 1$, but differ in their optimal trait value $\theta_j$. Since these resources all require specific adaptations to be effectively used and it is therefore impossible for individuals to specialize on multiple resources, we refer to these as the species-specific resources. For simplicity, we assume that all species-specific resources have the same maximum density in the absence of consumers ($R_{j,max} = R_{c,max}$; see electronic

supplementary material, appendix B for a relaxation of this assumption).

## (a) Parameterization of the feeding curves

For diversifying selection to emerge dynamically, directional selection must first drive a monomorphic population to a phenotype where ecological interactions induce disruptive selection [16]. In other words, the evolutionarily singular strategy should be an evolutionary branching point. In our model, where we explicitly take food dynamics into account, the shape and the location of the feeding curves (see equation (2.2)) strongly affect whether evolutionary branching occurs or not [18]. Since we aim to understand how ontogenetic diet shifts alter the conditions for speciation, we choose the width of the feeding curves (parameter $\tau$) and the distance between these curves ($|\theta_{i+1} - \theta_i|$) such, that in the absence of an ontogenetic diet shift ($m_{\text{shift}} = 0$), the ancestral species will radiate and its descendants colonize all available resources in the environment, resulting in six species where $x_i = \theta_i$. In contrast to [18], our model is size dependent and therefore we do not have an analytical expression for fitness. Hence, we cannot analytically express the conditions that will result in evolutionary branching. We therefore numerically calculate the distance between the feeding curves needed for evolutionary branching to occur given the parameters in electronic supplementary material, table A1. We find that for feeding curves with width $\tau = 1$, the distance between the curves should be between 1.9 and 4.2. Here, we choose a distance of 2.5 as the default value.

## (b) Model analysis of the deterministic model

We make use of the framework of adaptive dynamics [16] to study the evolution of the niche trait value $x_i$. Adaptive dynamics assumes that mutations are rare and only have small phenotypic effects. Since mutations occur infrequently, a successful variant reaches fixation before a new mutant arises. The ecological timescale is therefore considered much faster than the evolutionary timescale. All analyses were performed using the PSPManalysis software package [19]. This software package allows for the equilibrium and evolutionary analysis of physiologically structured population models (see [20,21] for more details). The model-specific files needed for the PSPManalysis together with R - scripts that execute all the calculations made in this article are available in the Dryad Digital Repository [22].

## (c) Individual-based model

The individual-based model is based on the same life history as the deterministic model described above (model details are described in the electronic supplementary material, appendix A). In the IBM, we use an additive diploid multi-locus genetic trait architecture. Each individual is assigned a genotype that in turn determines its phenotype. All individuals have two phenotypic traits, each of which is determined by $F_i$ ($i = x$ or $a$) diploid loci. One set of loci determines the ecological niche trait $x$, and the second set codes for the degree of assortative mating $a$. Assortative mating is based on the ecological niche trait $x$ and is described by a self-matching mate-choice function, following [4]. We assume that only females express their assortability, in keeping with female-limited mating. We assume full recombination among the loci. The niche trait $x$ is

determined by the sum of the values of its alleles; each allele can adopt every possible value. Since the assortative mating trait $a$ is bounded between −1 and 1, we assume that each allele can adopt a value between −1 and 1. The assortative mating trait $a$ is determined by the average value of all the alleles. For all simulations, we start with a monomorphic population where mating is random ($a = 0$). For computational purposes, we use a high mutation probability of $v = 0.1$ per allele. In the case of a mutation, the value of the offspring allele is drawn from a normal distribution with a mean equal to the parental value and a standard deviation of $\sigma = 0.01$. We implemented the model in C++, the code is available in the Dryad Digital Repository [22].

# 3. Results

## (a) Competition among small individuals hinders speciation

In case the productivity of the shared resource is low, an ancestral species will not diversify even though there is plenty of ecological opportunity (figure 1a,b). After colonization of the environment, the population evolves towards trait value $\theta_1$ and is therefore fully specialized on the first resource. This occurs both in clonal (figure 1a) and sexual (figure 1b) populations.

The result described above remains qualitatively the same when starting with an ancestral species with a different niche trait value $x_1$. The ancestral species will always specialize on one of the species-specific resources and there is no diversification (electronic supplementary material, figure B1 in appendix B).

The reason that the species fails to radiate in the presence of ecological opportunity is that competition for the species-specific resources is not strong enough. When the supply rate of the shared resource is low, newborn individuals have little of the shared resource available (grey line in figure 2a). They therefore grow slowly and shift to the other resources late in life (figure 2c). Because of the slow growth early in life, many individuals will die before they reach the body mass $m_{\text{shift}}$ where they can feed upon the species-specific resources. This results in a population with relatively few large individuals (black line in figure 2b). Since there are few large individuals, competition for the species-specific resources is weak and the density of these resources remains high (figure 2a). There is therefore no disruptive selection and individuals with a phenotype that deviates from the optimum in the resource niche $R_1$ fail to invade.

## (b) High food availability for small individuals results in adaptive radiation

An ancestral species will diversify to exploit all available resources in case of high productivity of the shared resource. Ultimately, there will be six different species that are each specialized to feed on one of the species-specific resources when large ($m \geq m_{\text{shift}}$; figure 1c,d). We find the adaptive radiation irrespective of the initial trait value of the ancestral species (electronic supplementary material, figure B2 in appendix B).

For high supply rates of the shared resource, newborn individuals have abundant resources available (grey line in figure 2a). They therefore grow fast and gain access to the

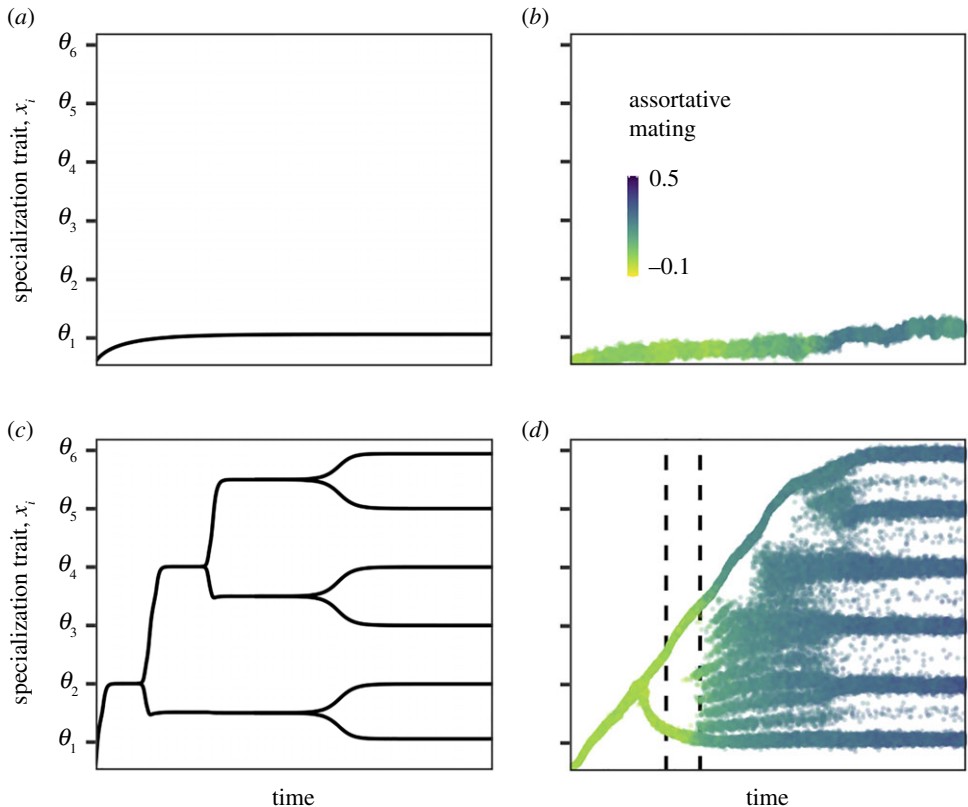

**Figure 1.** Evolutionary time series of the niche trait $x_i$ in case of (a,b) low and (c,d) high productivities of the shared resource; (a,c) are calculated using the adaptive dynamics approach and (b,d) are results from the IBM. Optimal trait values for feeding on the different species-specific resources are indicated by the ticks on the vertical axes. The colour of the dots in the (b,d) indicates the value of the assortative mating trait. Only mature individuals are plotted in (b,d). At the end of the simulation, reproductive isolation equals 0.97 in (d). At two points (indicated with vertical, dashed lines), we show in figure 3 the growth rate of the two morphs and their hybrid offspring. The ancestral population consists of a single species with a niche trait value of $x_1 = 0$. The supply rate of the shared resource equals $\delta R_{s,max} = 0.025$ g m$^{-3}$day$^{-1}$ in (a,b) and $\delta R_{s,max} = 0.2$ g m$^{-3}$day$^{-1}$ in (c,d). The body mass at birth $m_b$ equals 0.05 g, individuals shift to the species-specific resources at a body mass of $m_{shift} = 5$ g. The feeding specialization late in life does not affect feeding ability on the shared resource early in life ($\tau_s = \infty$). All other parameter values are as shown in the electronic supplementary material, tables A1 and A4. (Online version in colour.)

other resources early in life (figure 2d). Because of this fast growth early in life, many individuals will reach the body mass $m_{shift}$ where they can feed upon the species-specific resources. This results in a population with a high density of large individuals (black line in figure 2b). Since there are many large individuals, the depletion of the species-specific resources is strong, resulting in strong competition for these resources and subsequently diversification of the ancestral species.

When individuals reproduce sexually, we find that assortative mating evolves (figures 1d and 3c,d) and allows for the formation of six distinct species (figure 1d). Initially, mating is random (figure 3d and 1d). After a period of directional selection, the population splits into two morphs. The two morphs diverge from each other over time, one of the morphs evolves towards a niche trait value that is optimal to feed upon $R_1$, while the other morph evolves a niche trait value that is optimal to feed upon $R_3$. Mating is still random (figure 3c,d), which therefore results in many offspring with intermediate trait values. These individuals are best suited to feed upon $R_2$ and have less food available compared to individuals with more extreme trait values. Small differences in food availability may result in large fitness differences among individuals, since the effect of less food will accumulate over an individual's lifetime. Around the first diversification event, individuals with intermediate trait values will therefore grow slowly (figure 3a) and often die before they are large enough to reproduce. Because of this, we find for a short time

period two distinct morphs among adults, despite random mating (figure 3d) and plenty of gene flow (figure 3c).

Over time, the two morphs specialize more on separate resources. This releases competition for food for the individuals with intermediate trait values, allowing them to reach maturity (figure 3b). Because mating is still random, the two distinct morphs collapse into a hybrid swarm (second vertical line in figure 1d). Many of the offspring have intermediate phenotypes that are not suitable for efficient exploitation of the species-specific resources. This results in selection to evolve assortative mating (figure 3d). Once assortativeness is strong enough, the population splits into six ecologically different species which become almost completely reproductively isolated (RI = 0.96; figure 3c). Each of these species is specialized on one of the species-specific resources. Note that some of the successful species arise due to hybridization.

## (c) The faster individuals reach the size where ecological differentiation is possible, the more likely adaptive radiation is to occur

The earlier individuals have access to the species-specific resources, the lower the productivity of the shared resource may be and speciation still occurs (figure 4a,c). When individuals shift to the species-specific resources at a smaller body mass, they need to grow less before they are able to access

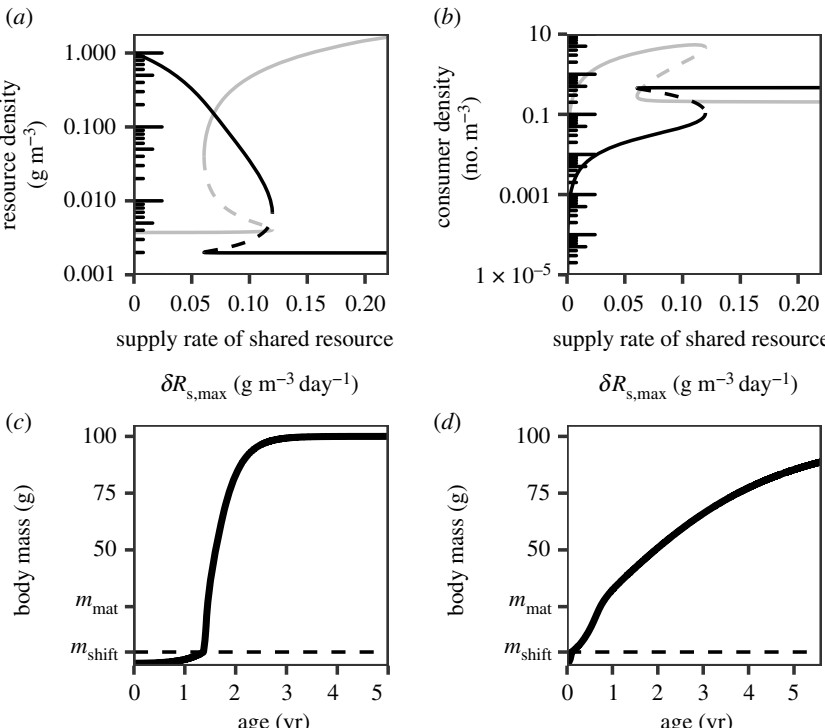

**Figure 2.** The productivity of the shared resource strongly affects the size-structure of the population and therefore the likelihood of speciation; (a) shows the density of the shared resource $R_s$ (grey) and resource $R_1$ (black) as a function of the supply rate of the shared resource; (b) shows the density of small consumers ($m < m_{shift}$) in grey, and the density of large consumers ($m \geq m_{shift}$) in black as a function of the supply rate of the shared resource. The solid lines represent stable ecological equilibria, the dashed lines represent unstable ecological equilibria. The consumers are, after the ontogenetic diet shift, specialized in feeding upon resource $R_1$ ($x_1 = 1$); (c,d) show individual growth of individuals specialized on $R_1$ ($x = 1$) in an environment with a (c) low or (d) high density of the shared resource. The horizontal dashed lines indicate the body mass at which individuals switch to the species-specific resources. The supply rate of the shared food source ($\delta R_{s,max}$) equals (c) 0.025 and (d) 0.2 g m$^{-3}$day$^{-1}$. The body mass at birth $m_b$ equals 0.05 g, individuals shift to the species-specific resources at a body mass of $m_{shift} = 5$ g. The feeding specialization late in life does not affect feeding ability on the shared resource earlier in life ($\tau_s = \infty$). All other parameter values are as shown in the electronic supplementary material, table A1. (Online version in colour.)

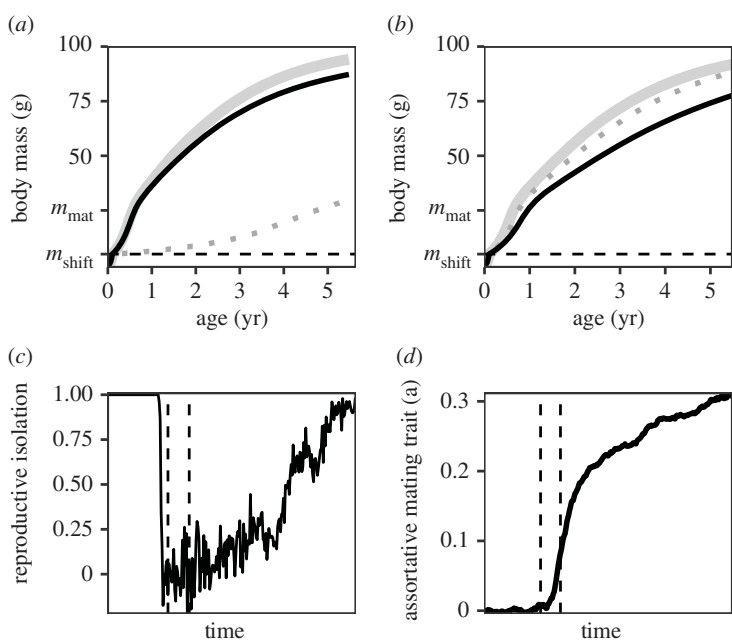

**Figure 3.** Growth curves of the two morphs (solid lines) and their hybrid offspring (dotted lines) just after the first diversification event (a) and just before the collapse of two morphs into a hybrid swarm (b); (c) shows the amount of reproductive isolation over evolutionary time (with a value of 0 indicating random mating and a value of 1 complete isolation), while (d) shows the average value of the assortative mating trait in the population. The vertical lines in these panels correspond to the growth curves in (a,b). In (a), hybrid offspring grow slowly and almost never mature (probability to reach maturity $<2 \times 10^{-8}$). The supply rate of the shared food source equals $\delta R_{s,max} = 0.2$ g m$^{-3}$day$^{-1}$. The body mass at birth $m_b$ equals 0.05 g, individuals shift to the species-specific resources at a body mass of $m_{shift} = 5$ g. In (a), the $R_1$ specialist (black line) has a trait value of $x = 1.57$, the hybrid $R_2$ specialist (dotted line) has a trait value of $x = 3.29$, and the $R_3$ specialist (grey line) has a trait value of $x = 4.87$. In (b), the $R_1$ specialist has a trait value of $x = 1.3$, the hybrid $R_2$ specialist has a trait value of $x = 3.83$ and the $R_3$ specialist has a trait value of $x = 6.61$. All other parameter values are as shown in the electronic supplementary material, table A1 and A4. (Online version in colour.)

the species-specific resources. Therefore, they switch to the species-specific resources earlier, thereby increasing the density of large individuals and subsequently competition for these resources. Hence, an adaptive radiation will happen even at a relatively low productivity rate of the shared resource when individuals can specialize on the species-specific resources early in their life.

Figure 4$b$,$d$ shows the number of species that will evolve as a function of the body mass at birth and the productivity of the shared resource. The larger individuals are at birth, the less they need to grow before they can use the species-specific resources. Therefore, an adaptive radiation can occur even at low productivity of the shared resource. When adults provide their young with food derived from the species-specific resources, as is the case in mammals and birds, for example, body mass at birth should be interpreted as the body mass at which offspring become independent from their parents.

Speciation is less likely to occur when there is a trade-off between the efficiency of feeding on the shared resource early in life and on a species-specific resource later in life (figure 4$c$,$d$). Due to the trade-off, individuals with a relatively high trait value $x_i$ will be less efficient as small individuals on the shared resource, resulting in a population with relatively few large individuals. Competition for the species-specific resources is therefore often not strong enough to result in disruptive selection. Only for high supply rates of the shared resource is the bottleneck among small individuals lifted and an adaptive radiation unfolds (figure 4$c$,$d$).

### (d) The occurrence of an adaptive radiation depends on the initial population composition

If an adaptive radiation occurs or not depends not only on abiotic conditions and the timing of ontogenetic niche shift, but also on the initial composition of the population. As is shown in figure 2, there is ecological bistability for intermediate supply rates of the shared resource. The population is either dominated by small or by large individuals. This ecological bistability has for a large range of parameter values consequences for the evolutionary dynamics. A population initially dominated by small individuals will not diversify over evolutionary time (figure 5$a$). On the other hand, a population dominated by large individuals experiences high intraspecific resource competition, which leads to disruptive selection and hence speciation (figure 5$b$). This evolutionary bistability is more common in the presence of a developmental trade-off between feeding on the shared resource and one of the species-specific resources (hatched areas in figure 4$c$,$d$).

### (e) Robustness of results

In electronic supplementary material, appendix B, we show that our results are robust against changes in model assumptions and parameters. In the main text, we assume an abrupt ontogenetic diet shift, such that individuals at any point in time either feed upon the shared resource or on the species-specific resources. In the electronic supplementary material, appendix B, we show that relaxing this assumption, such that there is a gradual diet shift, does not affect our results qualitatively (electronic supplementary material, figure B3). When we assume an ontogenetic diet broadening instead of a shift, such that large individuals continue to feed upon

the shared resource, our results do not change qualitatively either (electronic supplementary material, figure B4). However, in this case, speciation becomes more difficult, since competition among small individuals is intensified when large individuals continue to forage on the shared resource, resulting in less recruitment to larger size classes. We also show in electronic supplementary material, appendix B that relaxing our assumption that all species-specific resources have the same supply rate $\delta R_{c,max}$ does not change our results as long as the differences between the resources are not too large (electronic supplementary material, figure B5). In case one of the resources has a much lower supply rate compared to the others, no population will specialize on this resource and therefore fewer species will evolve. Results are also not affected by changes in body mass at maturation (electronic supplementary material, figure B6) or changes in background mortality rates (electronic supplementary material, figure B7).

## 4. Discussion

Here, we show that in species with an ontogenetic diet shift, competition among small individuals can hinder the process of adaptive radiation, even in the presence of plenty of ecological opportunity. Theory [4,5] and empirical evidence [23,24] show that by causing negative frequency-dependent interactions, intraspecific resource competition can be a source of disruptive selection and may contribute to speciation. However, in species with an ontogenetic diet shift, competition among small individuals results in resource scarcity and a developmental bottleneck that allows little recruitment to larger size-classes. Hence, there are only relatively few large individuals such that competition among them is too weak to result in disruptive selection.

We find that the initial size structure of the population determines if a radiation occurs or not. A population initially dominated by large individuals can often radiate and end up exploiting all available resources. On the other hand, a population dominated by small individuals will fail to fill all niches. This evolutionary bistability is a consequence of the ecological bistability between a population dominated by either small or large individuals, a common feature in models where individuals undergo ontogenetic diet shifts [25–27].

We show that the faster individuals reach the size where ecological differentiation is possible, the more likely it is that speciation happens. A short period on the shared resource reduces the magnitude of the developmental bottleneck among small individuals and therefore increases intraspecific competition among individuals large enough to specialize. This result implies that in species with parental care or high investment in offspring, an adaptive radiation is more likely to occur compared to similar species without parental care or with lesser parental investment. Theoretical and empirical studies show that increasing parental investment is advantageous when this results in offspring spending less time in relatively unfavourable habitats (e.g. [28–32]). We therefore expect parental care or larger eggs to evolve in the presence of a strong developmental bottleneck due to competition in the early life stages. These adaptations may reduce the time spend in early life stages and subsequently promote adaptive radiation by ecological specialization of the adults. Our model prediction could be tested by comparing speciation rates of related taxa that vary in egg size or the amount of parental care they provide.

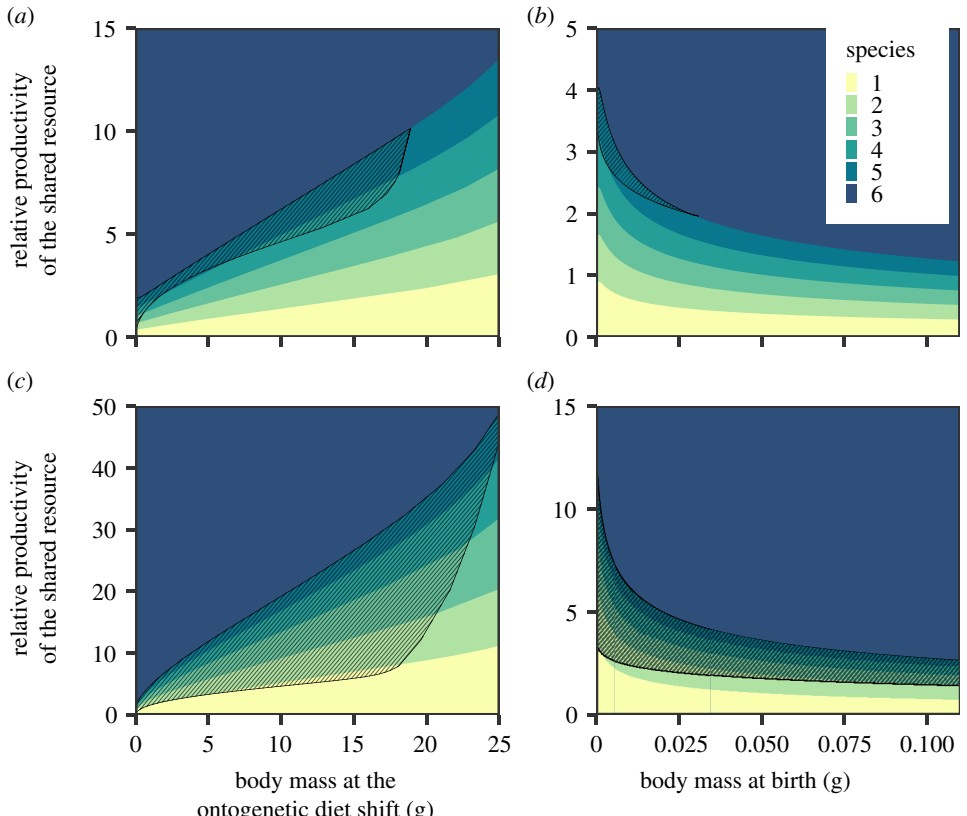

**Figure 4.** Two parameter plot showing the number of species that will evolve as a function of the relative productivity of the shared resource ($R_{s,max}/R_{c,max}$) and $m_{shift}$, the body mass at which individuals switch from the shared resource to the other resources (a,c), or $m_b$, the body mass at which individuals are born (b,d). The hatched areas indicate the parameter areas where there is eco-evolutionary bistability. In (a,b), there is no trade-off between feeding upon the shared resource and feeding upon the specific-specific resources ($\tau_s = \infty$). In (c,d), we assume a weak trade-off ($\tau_s = 20$). The body mass at birth $m_b$ equals 0.5 mg in (a,c). Individuals shift to the species-specific resources at a body mass of $m_{shift} = 5$ g in (b,d). All other parameter values are as shown in the electronic supplementary material, table A1. The results of these plots are calculated using the adaptive dynamics approach and therefore assume clonal reproduction. (Online version in colour.)

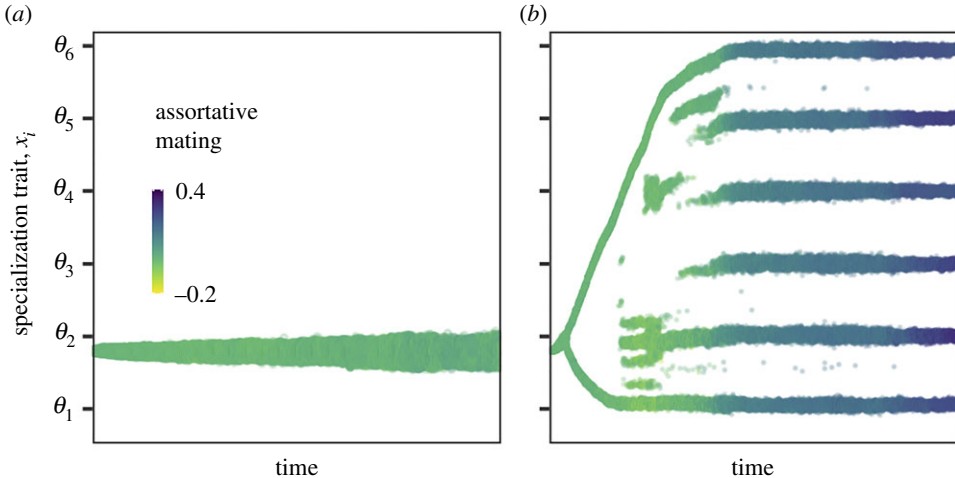

**Figure 5.** Evolutionary time series of the niche trait $x_i$ in case of a population initially dominated by small individuals (a) and a population initially dominated by large individuals (b). Optimal trait values for feeding on the different resources are indicated by the ticks on the vertical axes. The colour of the dots indicates the value of the assortative mating trait. Only mature individuals are plotted for clarity. At the end of the simulation, reproductive isolation equals 0.9 in (b). The ancestral population consists of a single species with a niche trait value of $x_1 = 3$. The supply rate of the shared resource equals $\delta R_{s,max} = 0.6$ g m$^{-3}$day$^{-1}$. The body mass at birth $m_b$ equals 0.5 mg, individuals shift to the species-specific resources at a body mass of $m_{shift} = 5$ g. There is a trade-off between feeding upon the shared resource and the species-specific resources ($\tau_s = 20$). The volume of the system equals $1 \times 10^3$ m$^3$ in (a) and $10^4$ m$^3$ in (b). All other parameter values are as shown in the electronic supplementary material, table A1 and A4. (Online version in colour.)

Cichlids provide some anecdotal evidence in support of our hypothesis. These fish have independently diversified within African lakes on multiple occasions and provide some of the most spectacular examples of adaptive radiation. The presence of female mouthbrooding is positively associated with the probability of adaptive radiation of cichlids

once they have colonized a lake [33]. In addition to providing extended parental care, mouthbrooders typically produce few large eggs [34]. These two factors both reduce ecological bottlenecks among small individuals and therefore promote speciation in our model. In contrast with our hypothesis, diversification rates are higher in marine gastropod species with small, long-lived, planktotrophic larvae compared with species that produce large, short-lived, lecithotrophic larvae [35]. One explanation for this discrepancy may be that developmental mode in marine invertebrates is correlated with dispersal ability. Dispersal results in less competition among offspring and in addition increases the extent of species ranges, and may thereby promote speciation, as long as dispersal rates are not too high (e.g. [36]).

The majority of all animal species undergo ontogenetic diet shifts [7]. Even in species that grow relatively little after birth, such as birds and mammals, niche shifts occur [7,37]. Because of their small size, newborn individuals are constrained in the type of food they can eat [38]. Larger individuals therefore often gain access to resources and habitats that they cannot use when small [7,39]. Although in many taxa and ecosystems species use similar resources when small and diverge in niche use when large (e.g. [9,40]), our assumption of a shared resource for the smallest individuals is not necessarily valid for all groups of animals. For example, in some spadefoot toad species, larvae can adopt two different feeding morphologies, but still produce the same adult morphology after metamorphosis [41]. When small individuals are able to specialize as well, competition among them can result in disruptive selection on the juvenile stage (e.g. [41]), which will considerably affect the conditions for an adaptive radiation. Thus, further research should consider the consequences of resource specialization in early life stages for adaptive radiations as well.

A clear avenue for further work is to combine the evolution of resource specialization with the evolution of traits that result in less competition among the smallest individuals. We found that when large individuals have access to multiple resources, and when food availability for large individuals is hence high, competition among offspring becomes intense. It is likely that severe resource limitation in the early phase of life will result in selection on traits that allow individuals to avoid this competition. For example, there may be selection to evolve differences in reproductive timing or nursery habitat, which will reduce competition among the smallest individuals. The evolution of parental care or increased egg size will also weaken competition and thereby promote speciation. In addition, developmental plasticity can be induced by parental care and possibly promote speciation (e.g. [42]). Alternatively, the timing of the ontogenetic diet shift may evolve as well, which in itself can result in speciation [6,43].

We found that even a weak developmental trade-off between feeding on the shared resource when little and adaptation to the species-specific resources later in life strongly diminished the likelihood of speciation. Since different food types often require different morphologies to be efficiently used, species with an ontogenetic diet shift may face a trade-off between performance early and late in life [38,44]. Because metamorphosis can break up developmental correlations between life stages [45], such trade-offs in performance are probably less severe in metamorphosing species compared to non-metamorphosing species. Nonetheless, metamorphosis is not a new beginning and constrains may persist across the metamorphic boundary (e.g. [46]). In addition, the loss of metamorphosis via direct development is associated with the production of larger offspring [32,47,48], which would, according to our model, promote speciation. Phylogenetic studies show mixed evidence for the role of metamorphosis on speciation rates. In insects, the most prominent increase in diversification rates is associated with the origin of complete metamorphosis [49,50]. There is also some evidence of a high rate of lineage accumulation after the re-evolution of metamorphosis in plethodontid salamanders [51]. Contrarily, the loss of an actively feeding larval stage in frogs is associated with higher evolutionary rates in traits associated with feeding specialization [52]. Further work could address the interaction between adaptive radiations and life cycle evolution.

Disruptive selection and subsequent speciation can arise from frequency-dependent interactions other than intraspecific resource competition. Other factors promoting divergence in populations include predation [12,53,54] and mutualistic interactions [10]. Since predation is often habitat- and size-specific (e.g. [11]), and can affect the size structure of the prey community (e.g. [55,56]), it would be especially interesting for further research to study the effect of predation on speciation dynamics in size-structured populations. We expect that predation can both hinder (e.g. [33]) and promote speciation (e.g. [12]), depending on the size-class that is affected by the predator.

Some of the most iconic examples of adaptive radiations occur in large lakes or on oceanic islands [1], where there is plenty of ecological opportunity. However, such habitats are no guarantee for a radiation to evolve. For example, while about 50 lineages of teleost fish established populations in Lake Victoria, only haplochromine cichlid fish radiated into several hundreds of species [57]. The findings of our study show that the timing of ontogenetic diet shifts strongly affects the conditions for an adaptive radiation in the presence of ecological opportunity. Individual development is a fundamental property of organisms, and the size of an individual does not only affect resource acquisition, but also other ecological interactions such as predation pressure, migration success and interspecific competition. Therefore, in order to understand the ecological conditions that promote adaptive radiations, it is necessary to take the full life cycle of individuals into account.

Data accessibility. Code and scripts used for this paper are available from the Dryad Digital Repository [22].

Authors' contributions. H.t.B.: conceptualization, formal analysis, investigation, writing—original draft and writing—review and editing; O.S.: writing—review and editing.

Both authors gave final approval for publication and agreed to be held accountable for the work performed therein.

Competing interests. We declare we have no competing interests.

Funding. H.t.B. was supported by an Eawag Fellowship.

Acknowledgements. We would like to thank the Eawag Fish Ecology and Evolution group for valuable discussions. Special thanks go to Catalina Chaparro-Pedraza, Anna Feller and Cameron Hudson for helpful comments on an earlier version of this paper.

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
