## [Peer Review File · Proceedings of the Royal Society B: Biological Sciences]

Review History

RSPB-2021-2655.R0 (Original submission)

Review form: Reviewer 1

Recommendation

Accept with minor revision (please list in comments)

Scientific importance: Is the manuscript an original and important contribution to its field?

Good

General interest: Is the paper of sufficient general interest?

Acceptable

Quality of the paper: Is the overall quality of the paper suitable?

Good

Is the length of the paper justified?

Yes

Should the paper be seen by a specialist statistical reviewer?

No

Do you have any concerns about statistical analyses in this paper? If so, please specify them explicitly in your report.

No

It is a condition of publication that authors make their supporting data, code and materials available - either as supplementary material or hosted in an external repository. Please rate, if applicable, the supporting data on the following criteria.

Is it accessible?

N/A

Is it clear?

N/A

Is it adequate?

N/A

Do you have any ethical concerns with this paper?

No

Comments to the Author

The paper deals with adaptive radiation of a consumer species undergoing diet shift during its complex life cycle. It emphasizes the importance of life history as a driver of complex eco-evolutionary dynamics. It is well written and presented, easy to read, and figures are of good quality and informative. I only have minor concerns that are attached. Main ones are to possibly include more model description in the Appendices, and expand the discussion of the results with respect to previous models (some suggestions are included in the attached comments)

Review form: Reviewer 2

Recommendation

Accept with minor revision (please list in comments)

Scientific importance: Is the manuscript an original and important contribution to its field?

Excellent

General interest: Is the paper of sufficient general interest?

Good

Quality of the paper: Is the overall quality of the paper suitable?

Good

Is the length of the paper justified?

Yes

Should the paper be seen by a specialist statistical reviewer?

Yes

Do you have any concerns about statistical analyses in this paper? If so, please specify them explicitly in your report.

No

It is a condition of publication that authors make their supporting data, code and materials available - either as supplementary material or hosted in an external repository. Please rate, if applicable, the supporting data on the following criteria.

Is it accessible?

Yes

Is it clear?

Yes

Is it adequate?

Yes

Do you have any ethical concerns with this paper?

No

Comments to the Author

In this paper the authors develop a model to explore how sympatric speciation may be influenced by ontogenetic shifts in resource extraction. The authors main conclusion is that high competition in early life increases mortality, thereby decreasing the density of adults and reducing competition for shared resources, reducing the chances of sympatric speciation by virtue of weak disruptive selection.

This ms is not directly in my field of work, and so I am not familiar with the modelling approach and won't comment on any technical details. That said, I liked this paper, and the fact that I (mostly) understood it - it being outside my research area - speaks to the clarity of writing and quality of scholarship. My sense is that the simplicity of the result must underlie some simplifying assumptions in the model, and while I think it is important to think critically about assumptions, I feel like the general result is interesting, broad, and apparently robust to relaxing some assumptions, and as such there may be little to gain in nit-picking the model details and the assumptions (although here I'd defer to an expert). In that spirit, I have a few broad comments about the authors assumptions and findings that may help frame the results and discussion.

One general feature this study led me to think about is the ecology of populations with different life histories. The authors equate large propagule size with larger adult population sizes, but in the most simplistic sense, a large propagule size also means lower per capita fecundity, and so potentially reduced population sizes of juveniles and adults. My general sense, which may or may not be true, is that species with low fecundity, large propagule sizes, and delayed maturation tend to be those with smaller population sizes to begin with. If true, then this may be at odds with one of the main inferences, which is that species with more parental investment/care (and generally "slower" life histories) tend to be more likely to experience large population sizes at adulthood, driving intraspecific competition and increasing the probability of disruptive selection. I don't think the authors should be required to perform a literature survey to investigate this, but if there are any readily available data or studies that investigate this, the authors could consider mentioning the nature of this relationship (if any).

Relatedly, life-histories evolve in response to the nature and timing of mortality. So any population experiencing strong juvenile competition and hence high juvenile mortality but low adult mortality may evolve to invest more in offspring, thereby decreasing juvenile mortality. The authors mention this at LN 286, but then the authors make the leap to suggest that parental care itself will lead to greater adult densities. All of this underlines that there may be something to gain in understanding (or at least speculate on) the relationship between life-history and adult population size.

Lastly, it would be interesting to have a quick look to see if there are any studies investigating diversification rate and the life-history spectrum among related taxa. A negative result does not

necessarily mean the authors thesis is false (e.g., there may be no ecological opportunity or limited diet shift, etc), but a positive result would be interesting and worth mentioning.

A few shorter comments:

The sentence LN268-272 could be clarified. "individuals shift to species-specific resources at a smaller body mass, they leave the smallest size class earlier, thereby increasing the density of large individuals and therefore competition for the species-specific resources." Is this because they grow faster after switching? If so, then I suggest rephrasing as "individuals shift to species-specific resources at a smaller body mass, they GROW MORE QUICKLY AND THEREFORE leave the smallest size class earlier, thereby increasing the density of large individuals and therefore competition for the species-specific resources". If my assumption is not correct, then please clarify how the result is realized.

Later in the discussion, the authors give a number of anecdotes that support their model. I'm sure there are anecdotes available that do not seem to be in accordance with the model. I think it would be worth pulling out a couple of "negative" anecdotes as well, as even if it is not clear "why" the results are discordant, other readers may make a connection (or at least better understand that not all evidence supports the model), and it may lead to further and unexpected insight into the model and the main finding.

If the authors need more room (word count) for editing, I suggest deleting the near-final paragraph of the ms about predation. This paragraph could be a manuscript (neigh, review ms; neigh, book) to itself, and is perhaps only worth mentioning very briefly, as an "its complicated, further study" note.

Decision letter (RSPB-2021-2655.R0)

17-Jan-2022

Dear Dr ten Brink,

Thank you for submitting this manuscript to Proceedings B. The manuscript has now been peer reviewed and the reviews have been assessed by an Associate Editor. The reviewers' comments (not including confidential comments to the Editor) and the comments from the Associate Editor are included at the end of this email for your reference. As you will see, the reviewers and the Editors have raised some concerns with your manuscript and we would like to invite you to revise your manuscript to address them.

When submitting your revision please upload a file under "Response to Referees" - in the "File Upload" section. This should document, point by point, how you have responded to the reviewers' and Editors' comments, and the adjustments you have made to the manuscript. We

require a copy of the manuscript with revisions made since the previous version marked as 'tracked changes' to be included in the 'response to referees' document.

Research ethics:

Use of animals and field studies:

It is a condition of publication that you make available the data and research materials supporting the results in the article. Please see our Data Sharing Policies (<https://royalsociety.org/journals/authors/author-guidelines/#data>). Datasets should be deposited in an appropriate publicly available repository and details of the associated accession number, link or DOI to the datasets must be included in the Data Accessibility section of the article (<https://royalsociety.org/journals/ethics-policies/data-sharing-mining/>). Reference(s) to datasets should also be included in the reference list of the article with DOIs (where available).

Online supplementary material will also carry the title and description provided during submission, so please ensure these are accurate and informative. Note that the Royal Society will not edit or typeset supplementary material and it will be hosted as provided. Please ensure that

the supplementary material includes the paper details (authors, title, journal name, article DOI). Your article DOI will be 10.1098/rspb.[paper ID in form xxxx.xxxx e.g. 10.1098/rspb.2016.0049].

Please submit a copy of your revised paper within three weeks. If we do not hear from you within this time your manuscript will be rejected. If you are unable to meet this deadline please let us know as soon as possible, as we may be able to grant a short extension.

We look forward to receiving your revision. If you have any questions at all, please do not hesitate to get in touch. Meanwhile, all the best for the New Year.

Best wishes,
Professor Loeske Kruuk
mailto:proceedingsb@royalsociety.org

Associate Editor
Board Member: 1
Comments to Author:

Here, the authors present a theoretical approach to understand how shifts in resource use during development can influence adaptive radiations, using adaptive dynamic and individual-based models. The manuscript has been assessed by two expert reviewers, and both are generally positive both about the approach and the clarity of the manuscript. On close reading, I agree: linking developmental processes with speciation is an exciting area of research that will be of interest to readers of Proceeding B.

The reviewers make a number of suggestions. Reviewer 1 raises several minor points for clarification (note that these are as comments/annotations on an attached PDF), suggests clearer incorporation of relevant theoretical studies, and requests further details on the model to be provided in the Appendix. Reviewer 2 raises broader discussion points about life-history variation linked to demography, e.g. how propagule size relates to adult population size, and the interaction between parental care or investment and density-dependent processes. By incorporating these suggestions (which range from the close detail through to broader discussion), the manuscript will be much improved both from the perspective of model detail and empirical relevance.

My own minor suggestions:

Abstract:

l23-24 'despite plenty of ecological opportunity': 'Ecological opportunity' not defined in abstract, perhaps here could just clarify that this relates to the adult stage (as it seems at odds with fact that the small individuals of this sentence are competing on shared resource, ie do not have ecological opportunity?).

Fig 1B,D - Seems only a small portion of the colour scale visible in the plot itself (mostly grey to pale green, so not much difference seen). Why does this need to be shaded from -1 through to 1 rather than the min to max values of the assortative mating trait?

l352 The theoretical work by Klug and Bonsall (Evol Ecol 2014, 2330-2351) on parental care and development rate is relevant here.

l379 You could expand the taxonomic breadth of empirical examples by linking this to studies on parental care, offspring competition and diversification in burying beetles, if deemed relevant (e.g. Jarrett et al 2018 Nat Comm 9,3987).

Reviewer(s)' Comments to Author:

Referee: 1

Comments to the Author(s)

The paper deals with adaptive radiation of a consumer species undergoing diet shift during its complex life cycle. It emphasizes the importance of life history as a driver of complex evolutionary dynamics. It is well written and presented, easy to read, and figures are of good quality and informative. I only have minor concerns that are attached. Main ones are to possibly

include more model description in the Appendices, and expand the discussion of the results with respect to previous models (some suggestions are included in the attached comments)

Referee: 2

Comments to the Author(s)

In this paper the authors develop a model to explore how sympatric speciation may be influenced by ontogenetic shifts in resource extraction. The authors main conclusion is that high competition in early life increases mortality, thereby decreasing the density of adults and reducing competition for shared resources, reducing the chances of sympatric speciation by virtue of weak disruptive selection.

This ms is not directly in my field of work, and so I am not familiar with the modelling approach and won't comment on any technical details. That said, I liked this paper, and the fact that I (mostly) understood it – it being outside my research area – speaks to the clarity of writing and quality of scholarship. My sense is that the simplicity of the result must underlie some simplifying assumptions in the model, and while I think it is important to think critically about assumptions, I feel like the general result is interesting, broad, and apparently robust to relaxing some assumptions, and as such there may be little to gain in nit-picking the model details and the assumptions (although here I'd defer to an expert). In that spirit, I have a few broad comments about the authors assumptions and findings that may help frame the results and discussion.

One general feature this study led me to think about is the ecology of populations with different life histories. The authors equate large propagule size with larger adult population sizes, but in the most simplistic sense, a large propagule size also means lower per capita fecundity, and so potentially reduced population sizes of juveniles and adults. My general sense, which may or may not be true, is that species with low fecundity, large propagule sizes, and delayed maturation tend to be those with smaller population sizes to begin with. If true, then this may be at odds with one of the main inferences, which is that species with more parental investment/care (and generally “slower” life histories) tend to be more likely to experience large population sizes at adulthood, driving intraspecific competition and increasing the probability of disruptive selection. I don't think the authors should be required to perform a literature survey to investigate this, but if there are any readily available data or studies that investigate this, the authors could consider mentioning the nature of this relationship (if any).

Relatedly, life-histories evolve in response to the nature and timing of mortality. So any population experiencing strong juvenile competition and hence high juvenile mortality but low adult mortality may evolve to invest more in offspring, thereby decreasing juvenile mortality. The authors mention this at LN 286, but then the authors make the leap to suggest that parental care itself will lead to greater adult densities. All of this underlines that there may be something to gain in understanding (or at least speculate on) the relationship between life-history and adult population size.

Lastly, it would be interesting to have a quick look to see if there are any studies investigating diversification rate and the life-history spectrum among related taxa. A negative result does not necessarily mean the authors thesis is false (e.g., there may be no ecological opportunity or limited diet shift, etc), but a positive result would be interesting and worth mentioning.

A few shorter comments:

The sentence LN268-272 could be clarified. “individuals shift to species-specific resources at a smaller body mass, they leave the smallest size class earlier, thereby increasing the density of large individuals and therefore competition for the species-specific resources.” Is this because they grow faster after switching? If so, then I suggest rephrasing as “individuals shift to species-specific resources at a smaller body mass, they **GROW MORE QUICKLY AND THEREFORE** leave the smallest size class earlier, thereby increasing the density of large individuals and

therefore competition for the species-specific resources". If my assumption is not correct, then please clarify how the result is realized.

Later in the discussion, the authors give a number of anecdotes that support their model. I'm sure there are anecdotes available that do not seem to be in accordance with the model. I think it would be worth pulling out a couple of "negative" anecdotes as well, as even if it is not clear "why" the results are discordant, other readers may make a connection (or at least better understand that not all evidence supports the model), and it may lead to further and unexpected insight into the model and the main finding.

If the authors need more room (word count) for editing, I suggest deleting the near-final paragraph of the ms about predation. This paragraph could be a manuscript (neigh, review ms; neigh, book) to itself, and is perhaps only worth mentioning very briefly, as an "its complicated, further study" note.

Author's Response to Decision Letter for (RSPB-2021-2655.R0)

See Appendix A.

RSPB-2021-2655.R1 (Revision)

Review form: Reviewer 1

Recommendation

Accept as is

Scientific importance: Is the manuscript an original and important contribution to its field?

Good

General interest: Is the paper of sufficient general interest?

Good

Quality of the paper: Is the overall quality of the paper suitable?

Good

Is the length of the paper justified?

Yes

Should the paper be seen by a specialist statistical reviewer?

No

Do you have any concerns about statistical analyses in this paper? If so, please specify them explicitly in your report.

No

It is a condition of publication that authors make their supporting data, code and materials available - either as supplementary material or hosted in an external repository. Please rate, if applicable, the supporting data on the following criteria.

Is it accessible?

Yes

Is it clear?

Yes

Is it adequate?

Yes

Do you have any ethical concerns with this paper?

No

Comments to the Author

The Authors thoroughly and thoughtfully addressed all concerns. I endorse publication

Review form: Reviewer 2

Recommendation

Accept as is

Scientific importance: Is the manuscript an original and important contribution to its field?

Excellent

General interest: Is the paper of sufficient general interest?

Excellent

Quality of the paper: Is the overall quality of the paper suitable?

Excellent

Is the length of the paper justified?

Yes

Should the paper be seen by a specialist statistical reviewer?

No

Do you have any concerns about statistical analyses in this paper? If so, please specify them explicitly in your report.

No

It is a condition of publication that authors make their supporting data, code and materials available - either as supplementary material or hosted in an external repository. Please rate, if applicable, the supporting data on the following criteria.

Is it accessible?

Yes

Is it clear?

N/A

Is it adequate?

N/A

Do you have any ethical concerns with this paper?

No

Comments to the Author

I think the authors have done a good job revising this work. They have made a clear effort to address and incorporate all the reviewer suggestions, even those of mine that missed the mark. Congratulations on this work - I think it will attract a broad readership and I look forward to seeing it in print!

Decision letter (RSPB-2021-2655.R1)

28-Feb-2022

Dear Dr ten Brink,

I am very pleased to inform you that your manuscript entitled "Competition among small individuals hinders adaptive radiation despite ecological opportunity" has been accepted for publication in Proceedings B.

Data Accessibility section

Open Access

Paper charges

Sincerely,
Professor Loeske Kruuk
Editor, Proceedings B
mailto: proceedingsb@royalsociety.org

Associate Editor:

Comments to Author:

The revised version has been assessed by the original expert reviewers, who are satisfied that the authors have incorporated all their changes. I agree, and I believe this paper, which links developmental competition with adaptive radiation, will be of interest to readers of Proceedings B. I only have three very minor suggestions on phrasing:

l24 "In case small individuals" -> 'In cases where small individuals...' or, more simply 'When small individuals...'

l56 Would move 'Also' to later in sentence ('are *also* a common phenomenon')

l140 "Here, we chose" -> "Here, we choose" (consistency of tense)

Appendix A

February 3, 2022

Dear Editor,

We're delighted that our manuscript 'Competition among small individuals hinders adaptive radiation despite ecological opportunity' (RSPB-2021-2655) was so favourably reviewed. We have used the comments of the first reviewer to improve the model description. Based on comments of both reviewers and the associate editor we have broadened the discussion and included a few more examples where possible. In order to not increase the length of the manuscript, we have shortened the paragraph about predation, following the suggestion of the second reviewer. Please see below for our detailed response to the comments, with the italic *Times New Roman* font containing our responses and the Arial font showing the original comments.

We hope that you will find the revised version improved and publishable in Proceedings B.

On behalf of both co-authors,

Hanna ten Brink

17-Jan-2022

Dear Dr ten Brink,

Thank you for submitting this manuscript to Proceedings B. The manuscript has now been peer reviewed and the reviews have been assessed by an Associate Editor. The reviewers' comments (not including confidential comments to the Editor) and the comments from the Associate Editor are included at the end of this email for your reference. As you will see, the reviewers and the Editors have raised some concerns with your manuscript and we would like to invite you to revise your manuscript to address them.

When submitting your revision please upload a file under "Response to Referees" - in the "File Upload" section. This should document, point by point, how you have responded to the reviewers' and Editors' comments, and the adjustments you have

made to the manuscript. We require a copy of the manuscript with revisions made since the previous version marked as 'tracked changes' to be included in the 'response to referees' document.

Research ethics:

Use of animals and field studies:

It is a condition of publication that you make available the data and research materials supporting the results in the article. Please see our Data Sharing Policies (<https://royalsociety.org/journals/authors/author-guidelines/#data>). Datasets should be deposited in an appropriate publicly available repository and details of the associated accession number, link or DOI to the datasets must be included in the Data Accessibility section of the article (<https://royalsociety.org/journals/ethics-policies/data-sharing-mining/>). Reference(s) to datasets should also be included in the reference list of the article with DOIs (where available).

Please submit a copy of your revised paper within three weeks. If we do not hear from you within this time your manuscript will be rejected. If you are unable to meet this deadline please let us know as soon as possible, as we may be able to grant a short extension.

We look forward to receiving your revision. If you have any questions at all, please do not hesitate to get in touch. Meanwhile, all the best for the New Year.

Best wishes,

Professor Loeske Kruuk
mailto: proceedingsb@royalsociety.org

Detailed response to reviewers

We would like to thank the associate editor and the reviewers for their time and useful suggestions to improve the manuscript. Below, we respond point to point to all comments. We use italic Times new Roman for our response, the original comments are in the Arial font. In case the reviewers would like to see the code & scripts we used for all calculations in the manuscript, they can download it at the link below.

https://datadryad.org/stash/share/RBsQ1SC15hIJvHx1yd8Wk7zUp6UR_jW1jxUvuey0djk

Associate Editor

Board Member: 1

Comments to Author:

Here, the authors present a theoretical approach to understand how shifts in resource use during development can influence adaptive radiations, using adaptive dynamic and individual-based models. The manuscript has been assessed by two expert reviewers, and both are generally positive both about the approach and the clarity of the manuscript. On close reading, I agree: linking developmental processes with speciation is an exciting area of research that will be of interest to readers of Proceeding B.

The reviewers make a number of suggestions. Reviewer 1 raises several minor points for clarification (note that these are as comments/annotations on an attached PDF), suggests clearer incorporation of relevant theoretical studies, and requests further details on the model to be provided in the Appendix. Reviewer 2 raises broader discussion points about life-history variation linked to demography, e.g. how propagule size relates to adult population size, and the interaction between parental care or investment and density-dependent processes. By incorporating these suggestions (which range from the close detail through to broader discussion), the manuscript will be much improved both from the perspective of model detail and empirical relevance.

My own minor suggestions:

AE.1: Abstract:

l23-24 'despite plenty of ecological opportunity': 'Ecological opportunity' not defined in abstract, perhaps here could just clarify that this relates to the adult stage (as it seems at odds with fact that the small individuals of this sentence are competing on shared resource, ie do not have ecological opportunity?).

Thank you for this suggestion, we now write 'despite plenty of ecological opportunity for large individuals' in the abstract (L23/L24 in both the track-changes and clean documents) to clarify.

AE.2: Fig 1B,D - Seems only a small portion of the colour scale visible in the plot itself (mostly grey to pale green, so not much difference seen). Why does this need to be shaded from -1 through to 1 rather than the min to max values of the assortative mating trait?

Apologies for the bad color scaling in our first version of the manuscript. We have now changed the color scale of figure 1 and figure 5 such that the evolutionary change in the assortative mating trait is easier to see.

AE.3: I352 The theoretical work by Klug and Bonsall (Evol Ecol 2014, 2330-2351) on parental care and development rate is relevant here.

Thank you for the suggestion. We now cite this work (ref 27) in the discussion as follows:

'Theoretical and empirical studies show that increasing parental investment is advantageous when this results in offspring spending less time in relatively unfavorable habitats (e.g., [27–31]). We therefore expect parental care or larger eggs to evolve in the presence of a strong developmental bottleneck due to competition in the early life stages. These adaptations may reduce the time spend in early life stages and subsequently promote adaptive radiation by ecological specialization of the adults.'

(L371-L376 in the track-changes document, and L300-L305 in the clean document)

AE.4: I379 You could expand the taxonomic breadth of empirical examples by linking this to studies on parental care, offspring competition and diversification in burying beetles, if deemed relevant (e.g. Jarrett et al 2018 Nat Comm 9,3987).

We have included another study of burying beetles in the discussion (Schrader et al 2021, Evolution, ref 41). The text now reads:

'For example, there may be selection to evolve differences in reproductive timing or nursery habitat, which will reduce competition among the smallest individuals. Evolution of parental care or increased egg size will also weaken competition and thereby promote speciation. In addition, developmental plasticity can be induced by parental care and possibly promote speciation (e.g., [41]).'

(L409-412 in the track-changes document, and L338-L341 in the clean document)

Reviewer(s)' Comments to Author:

Referee: 1

Comments to the Author(s)

The paper deals with adaptive radiation of a consumer species undergoing diet shift during its complex life cycle. It emphasizes the importance of life history as a driver of complex eco-evolutionary dynamics. It is well written and presented, easy to read, and figures are of good quality and informative. I only have minor concerns that are attached. Main ones are to possibly include more model description in the Appendices, and expand the discussion of the results with respect to previous models (some suggestions are included in the attached comments)

Thank you for your kind words and your useful suggestions. Below, we have copied your comments on the manuscript and shortly replied to each of them.

L52: but see

Variability in life-history switch points across and within populations explained by Adaptive Dynamics

P Landi, JR Vonesh, C Hui

Journal of the Royal Society Interface 15 (148), 20180371

Apologies for missing this relevant and interesting paper! We now cite this work (ref 6) at two places in the manuscript.

(L53 and L414 in the track-changes document, and L52, L343 in the clean document)

L62: also size-dependent artificial selection

Fisheries-induced disruptive selection

P Landi, C Hui, U Dieckmann

Journal of Theoretical Biology 365, 204-216

Thank you for the suggestion. Now cited (ref 13) on L64-L65 in the track-changes document, and L65-L66 in the clean document.

L110: should also be m_i ?

No, since body mass is not a fixed property of each species, but different per individual.

Therefore, the subscript i is not necessary. Also, the original paper where our model is based upon (Hartvig et al 2011), is not using the subscript i to indicate individual body mass.

L118: (x_i close to θ_s)?

We have added a short sentence to clarify: '(i.e., when x_i is close to θ_s)'

(L136 in the track-changes document, and L135 in the clean document)

L128: ESS is usually used for Evolutionarily Stable Strategy, which is the opposite of a branching point.

Although ESS is sometimes used for evolutionarily singular strategy in the adaptive dynamics literature, we agree with the reviewer that using this abbreviation will by many readers be interpreted as evolutionarily stable strategy. We therefore removed this abbreviation to avoid confusion.

(L129 in the track-changes document, and L128 in the clean document)

L135: with exactly 6 morphs with $x_i = \theta_i$?

We have clarified on L136 (L135 in the clean document). We now write: 'the ancestral species will radiate and its descendants colonize all available resources in the environment, resulting in six species where $x_i = \theta_i$ '

L137: even for the population deterministic model? why?

Our model is a size-structured model, and formulated in terms of partial differential equations. Even in a simplified case with only 2 species-specific resources, R_0 will depend on 3 resource densities that themselves depend on the density and size-structure of the consumer population. Even though it might be possible to provide a formal expression for R_0 , such an expression would not be useful at all because it will be impossible to solve analytically, since it is not possible to get analytical expressions for the resource densities. For an example of such a formal expression for R_0 in a size-structured model, see Chaparro-Pedraza & de Roos 2020 (doi 10.1111/evo.13957).

We now write in the manuscript:

'In contrast to [18], our model is size-dependent and therefore we do not have an analytical expression for fitness.'

(L137 in the track-changes document, L136 in the clean document)

L149: is this small enough for eco and evo timescale separation?

Even though the mutation rate is very high, the ecological and evolutionary dynamics of the IBM are comparable to the deterministic approach with a full separation of time scales. Our results seem to be robust against the separation of timescale assumption. Although this in itself is an interesting finding, we think that a discussion of this finding is not relevant for the current work.

L236: in the initial phase just after evolutionary branching?

We are not exactly sure about what the reviewer wants to ask. Just after evolutionary branching intermediate offspring are less fit, but we are not sure if this is what the reviewer asks for? Apologies for not being able to reply better to this comment.

L265: smaller the size where ecological differentiation is possible?

Not necessarily smaller, since when individuals are born at a large size, this will already promote speciation without a change in the size where ecological differentiation is possible. We explain this in the ms as follows (L282 - L285 in the track changes document and L237 - L240 in the clean document):

'The larger individuals are at birth, the less they need to grow before they can utilize the species-specific resources. Therefore, an adaptive radiation can occur even at low productivity of the shared resource.'

L267: smaller?

See our reply above, also when individuals are born larger speciation occurs more easily.

L268: for speciation to still occur?

Thank you for the suggestion, but we prefer the current formulation.

L287: ecological? evolutionary? or both?

To clarify, we have changed the subheading to 'The occurrence of an adaptive radiation depends on the initial population composition' (L95 in the track-changes document, and L250 in the clean document)

L302 (Figure 4): ecological? eco-evolutionary? is it driven by multiple equilibria in the ecological system, depending on ecological initial conditions? or driven by multiple evo equilibria depending on evo initial conditions?

Good point! It is driven by multiple ecological equilibria, but ecological bistability does not always result in evolutionary bistability. Therefore, we are now more careful in our wording in the text and write 'This ecological bistability has for a large range of parameter values consequences for the evolutionary dynamics', to indicate that the ecological bistability does not always result in evolutionary bistability (L300 in the track-changes document, and L255 in the clean document). In addition, we write in the caption of figure 4 "The hatched areas indicate the parameter areas where there is eco-evolutionary bistability"

Figure 4: other comments:

We now include the mathematical notation of the two parameters of interest in the caption of figure 4.

L330: thus reducing density of large individuals? it seems density-dependent competition among large individual is the main driver of diversification here

Thank you for this suggestion. We now write 'However, in this case speciation becomes more difficult, since competition among small individuals is intensified when large individuals continue to forage on the shared resource, resulting in less recruitment to larger size classes.'

(L342 in the track-changes document, and L269-271 in the clean document)

L343: but see

Variability in life-history switch points across and within populations explained by Adaptive Dynamics

P Landi, JR Vonesh, C Hui

Journal of the Royal Society Interface 15 (148), 20180371

L388: also Variability in life-history switch points across and within populations explained by Adaptive Dynamics

P Landi, JR Vonesh, C Hui

Journal of the Royal Society Interface 15 (148), 20180371

We now cite this nice paper at several places in the manuscript (Ref 6, L53 and L414 in the track-changes document, and L52, L343 in the clean document)

Language editing on L404 and L407:

We have changed this. (L433 in the track-changes document, and L359 in the clean document)

L408: (intraspecific competition is also frequency/density dependent)

We are aware that intraspecific competition is also frequency/density dependent, we write in the manuscript (L436 in the track-changes document, and L362 in the clean document) 'Disruptive selection and subsequent speciation can arise from frequency-dependent interactions other than intraspecific resource competition.'

L408 see also

Branching scenarios in eco-evolutionary prey-predator models

P Landi, F Dercole, S Rinaldi

SIAM Journal on Applied Mathematics 73 (4), 1634-1658

We now cite this research (ref 53, L438 in the track-changes document, and L364 in the clean document)

At several places in the ms the reviewer suggested to change gram to g.

Thanks for the suggestion, we prefer to keep using gram instead of g for clarification.

Appendix

L1: for the IBM, population model, or both?

Thanks for pointing this out! We now write in the table caption 'Model variables, parameters and their default value from [17] for both the deterministic and individual based model.'

Table A2: these and those in table A3 should also have a brief description/explanation

In the appendix, we now shortly describe our model. We hope that by doing so, the model details are clear to the reader.

is this a functional response?

We have updated the function description in table A2. It now reads:

Total food intake (Holling type 2 functional response for multiple prey)

We now also describe that we use a type 2 functional response in the model description (L20)

L9: population?

We now write 'In the deterministic model, the dynamics of the system involve consumer densities'

(L49)

EqA1: is this a density or a mass?

This is a density, we write 'the change in resource density' on L54

L14: S is only dividing this?

Yes, since we track only the densities of the resources, we do not translate this to individuals. Therefore, the supply term ($\Delta R_{j,max} - R_j$) has [$\text{gram m}^{-3} \text{day}^{-1}$] as its unit. The intake of resources by consumers (gram day^{-1}) should therefore be divided by the total size of the system to also result in density. We have added the following sentence to clarify:

'We divide this intake by the volume S to calculate the change in resource density due to consumption.'

See L56-L57

EqA3: why a_f is squared? a figure representing this function would be helpful

There is not a particular reason to use a_f^2 . We used the same mating function as has been used in the classical Nature paper of Dieckmann & Doebeli (1999), they also use the squared mating trait in their function. We now cite this paper in the appendix as well to clarify we use the same function. In addition, we have added a figure (fig A1)

L41: since $\theta_{i+1} - \theta_i = 2.5$, see table A1

Thank you for this suggestion, we now clarify this on L78 as follows:

'When the difference between the two traits is larger than $\frac{1}{4}(|\theta_{i+1} - \theta_i|)$ (which equals 0.625 for our choice of parameters, see table A1)'

L44: ($a=0$)?

We have included this L86

L57: produce

L100

Referee: 2

Comments to the Author(s)

In this paper the authors develop a model to explore how sympatric speciation may be influenced by ontogenetic shifts in resource extraction. The authors main conclusion is that high competition in early life increases mortality, thereby decreasing the density of adults and reducing competition for shared resources, reducing the chances of sympatric speciation by virtue of weak disruptive selection.

This ms is not directly in my field of work, and so I am not familiar with the modelling approach and won't comment on any technical details. That said, I liked this paper, and the fact that I (mostly) understood it – it being outside my research area – speaks to the clarity of writing and quality of scholarship. My sense is that the simplicity of the result must underlie some simplifying assumptions in the model, and while I think it is important to think critically about assumptions, I feel like the general result is interesting, broad, and apparently robust to relaxing some assumptions, and as such there may be little to gain in nit-picking the model details and the

assumptions (although here I'd defer to an expert). In that spirit, I have a few broad comments about the authors assumptions and findings that may help frame the results and discussion.

We are very pleased to read this positive evaluation! Thank you for your suggestions to improve the manuscript.

R2.1: One general feature this study led me to think about is the ecology of populations with different life histories. The authors equate large propagule size with larger adult population sizes, but in the most simplistic sense, a large propagule size also means lower per capita fecundity, and so potentially reduced population sizes of juveniles and adults. My general sense, which may or may not be true, is that species with low fecundity, large propagule sizes, and delayed maturation tend to be those with smaller population sizes to begin with. If true, then this may be at odds with one of the main inferences, which is that species with more parental investment/care (and generally "slower" life histories) tend to be more likely to experience large population sizes at adulthood, driving intraspecific competition and increasing the probability of disruptive selection. I don't think the authors should be required to perform a literature survey to investigate this, but if there are any readily available data or studies that investigate this, the authors could consider mentioning the nature of this relationship (if any).

In our model, we don't fix adult population size, this is an emergent property. We find that larger propagule size results in stronger competition among adults (via increased densities) because larger propagule sizes result in a shorter juvenile period and higher recruitment to the adult stage (explained on L282-283 in the track changes document and L238-L239 in the clean document). We agree with the reviewer that, everything else equal, larger propagule size also results in fewer offspring (because of the size versus number of offspring trade-off, see equation A6 in the supplementary materials, the number of offspring an individual produces decreases linearly with the size of their offspring, parameter m_b), but because of the above-mentioned feedback loop, this not necessarily translates into a smaller total population size.

In addition, it is not necessarily adult population size that matters for the adaptive radiation, but the strength of competition among the adults/large individuals. Therefore, what our model predicts is that in species with parental care and/or larger offspring, there is less competition early in life, resulting in more competition later in life (explained on L273-L280 in the track changes document and L228-L235 in the clean document:

'When individuals shift to the species-specific resources at a smaller body mass, they need to grow less before they are able to access the species-specific resources. Therefore, they switch to the species-specific resources earlier, thereby increasing the density of large individuals and subsequently competition for these resources.'

That said, the suggestion of the reviewer to look at studies on the relation between population size and propagule size is potentially interesting, especially when such studies also investigate the size-structure of the populations, since the (relative) population size of adults is what matters.

We are not aware of studies showing that larger offspring size subsequently result in larger adult population sizes. However, there are plenty of studies showing that within species, strong competition and high population densities selects for more parental investment.

We now refer to these findings in the manuscript as follows:

'Theoretical and empirical studies show that increasing parental investment is advantageous when this results in offspring spending less time in relatively unfavorable habitats (e.g., [27–31]). We therefore expect parental care or larger eggs to evolve in the presence of a strong developmental bottleneck due to competition in the early life stages. These adaptations may reduce the time spend in early life stages and subsequently promote adaptive radiation by ecological specialization of the adults.'

(L371-L376 in the track-changes document and L300-L305 in the clean document)

R2.2: Relatedly, life-histories evolve in response to the nature and timing of mortality. So any population experiencing strong juvenile competition and hence high juvenile mortality but low adult mortality may evolve to invest more in offspring, thereby decreasing juvenile mortality. The authors mention this at LN 286, but then the authors make the leap to suggest that parental care itself will lead to greater adult densities. All of this underlines that there may be something to gain in understanding (or at least speculate on) the relationship between life-history and adult population size.

See our reply above to R2.1.

R2.3: Lastly, it would be interesting to have a quick look to see if there are any studies investigating diversification rate and the life-history spectrum among related taxa. A negative result does not necessarily mean the authors thesis is false (e.g., there may be no ecological opportunity or limited diet shift, etc), but a positive result would be interesting and worth mentioning.

In our manuscript, we explicitly look at the effect of (the timing of) ontogenetic diet shifts, size at birth, and size-specific interaction on diversification and speciation. Of course, there will be other aspects of life-history contributing to diversification and speciation, e.g., developmental mode (which we discuss on L419-435 in the track-changes document and L345-L361 in the clean document) or sexual dimorphism (see for example de Lisle et al, 2015 (10.1098/rspb.2014.2213)), but that is not the focus of our work. Therefore, we think it is relevant to look at empirical studies on diversification in relation to the factors that we looked at in our model (diet shifts, size at birth), while other life-history aspects are not relevant for the current study.

We have extensively searched for empirical studies on the relationship between development, ontogenetic diet shifts, parental investment, and speciation/diversification rates.

Unfortunately, there are only a few studies, many of them we cite already in the manuscript. Based upon a suggestion by the associate editor (AE.4) and your comments, we have added a few more examples of how life-history affects speciation.

On L413-414 (L342-343 in the clean document) we shortly link developmental plasticity, parental care, and speciation. We write: 'In addition, developmental plasticity can be induced by parental care and possibly promote speciation (e.g., [41]).'

On L379 - 391 (L308-L320 in the clean document) we discuss several studies on the effect of parental investment on diversification, one example confirming our hypotheses, the other one in contradiction with our hypothesis. In addition, we shortly link diversification to dispersal. This paragraph now reads:

'Cichlids provide some anecdotal evidence in support of our hypothesis. These fish have independently diversified within African lakes on multiple occasions and provide some of the most spectacular examples of adaptive radiation. The presence of female mouthbrooding is positively associated with the probability of adaptive radiation of cichlids once they have colonized a lake [32]. In addition to providing extended parental care, mouthbrooders typically produce few large eggs [33]. These two factors both reduce ecological bottlenecks among small individuals and therefore promote speciation in our model. In contrast to our hypothesis, diversification rates are higher in marine gastropod species with small, long-lived, planktotrophic larvae compared to species that produce large, short-lived, lecithotrophic larvae [34]. One explanation for this discrepancy may be that developmental mode in marine invertebrates is correlated with dispersal ability. Dispersal results in less competition among offspring and in addition increases the extent of species ranges and may thereby, as long as dispersal rates are not too high (e.g. [35]), promote speciation.'

A few shorter comments:

R2.4: The sentence LN268-272 could be clarified. "individuals shift to species-specific resources at a smaller body mass, they leave the smallest size class earlier, thereby increasing the density of large individuals and therefore competition for the species-specific resources." Is this because they grow faster after switching? If so, then I suggest rephrasing as "individuals shift to species-specific resources at a smaller body mass, they GROW MORE QUICKLY AND THEREFORE leave the smallest size class earlier, thereby increasing the density of large individuals and therefore competition for the species-specific resources". If my assumption is not correct, then please clarify how the result is realized.

Thank you for pointing out that this part of the ms is unclear. The reason that they leave the smallest size class earlier, is that they have to grow less in body mass. We have rewritten these sentences to clarify. The text now reads:

'When individuals shift to the species-specific resources at a smaller body mass, they need to grow less before they are able to access the species-specific resources. Therefore, they switch to the species-specific resources earlier, thereby increasing the density of large individuals and subsequently competition for these resources.'

(L274-278 in the track changes document and on L229-L233 in the clean document)

R2.5: Later in the discussion, the authors give a number of anecdotes that support their model. I'm sure there are anecdotes available that do not seem to be in accordance with the model. I think it would be worth pulling out a couple of "negative" anecdotes as well, as even if it is not clear "why" the results are

discordant, other readers may make a connection (or at least better understand that not all evidence supports the model), and it may lead to further and unexpected insight into the model and the main finding.

We agree that it is useful to add examples that are in contrast with what our model predicts. Based upon your other comment (R2.3), we have been able to find another study in contrast to our results (see our reply to R2.3). Unfortunately, we were not able to find yet other relevant studies.

R2.6: If the authors need more room (word count) for editing, I suggest deleting the near-final paragraph of the ms about predation. This paragraph could be a manuscript (neigh, review ms; neigh, book) to itself, and is perhaps only worth mentioning very briefly, as an “its complicated, further study” note.

We appreciate your suggestion of shortening this paragraph. We decided to remove this paragraph completely. We have added a single sentence to the previous paragraph to shortly explain our expectation of how predation will affect the results

'We expect that predation can both hinder (e.g. [32]) and promote speciation (e.g. [12]), depending on the size-class that is affected by the predator.'

(L441-L443 in the track changes document, and L367 - L369 in the clean document)